# Unsupervised Semantic Aggregation and Deformable Template Matching for Semi-Supervised Learning

**Tao Han**[†] **Junyu Gao**[†] **Yuan Yuan and Qi Wang**[*]
School of Computer Science and Center for OPTical IMagery Analysis and Learning
Northwestern Polytechnical University
Xi'an, Shaanxi, P.R. China.
hantao10200@mail.nwpu.edu.cn, {gjy3035, y.yuan1.ieee, crabwq}@gmail.com

## Abstract

Unlabeled data learning has attracted considerable attention recently. However, it is still elusive to extract the expected high-level semantic feature with mere unsupervised learning. In the meantime, semi-supervised learning (SSL) demonstrates a promising future in leveraging few samples. In this paper, we combine both to propose an Unsupervised Semantic Aggregation and Deformable Template Matching (USADTM) framework for SSL, which strives to improve the classification performance with few labeled data and then reduce the cost in data annotating. Specifically, unsupervised semantic aggregation based on Triplet Mutual Information (T-MI) loss is explored to generate semantic labels for unlabeled data. Then the semantic labels are aligned to the actual class by the supervision of labeled data. Furthermore, a feature pool that stores the labeled samples is dynamically updated to assign proxy labels for unlabeled data, which are used as targets for cross-entropy minimization. Extensive experiments and analysis across four standard semi-supervised learning benchmarks validate that USADTM achieves top performance (e.g., 90.46% accuracy on CIFAR-10 with 40 labels and 95.20% accuracy with 250 labels). The code is released at https://github.com/taohan10200/USADTM.

## 1 Introduction

Deep learning is booming driven by massive labeled data over the past few years, such as image classification [1, 2], semantic segmentation [3, 4], object detection [5, 6], natural language processing [7, 8]. Besides, learning with unlabeled data also makes much progress in reducing the labeling costs [9–11]. The two most important branches are unsupervised and semi-supervised learning. For the image classification task, semi-supervised learning has shown that it can achieve a performance close to supervised results under certain conditions, while unsupervised learning remains a huge challenge for machine learning. A new perspective is to combine unsupervised learning with supervised learning to achieve better performance.

In this paper, we are working on the following problems. 1) Most of the unsupervised methods cannot directly output the classification results of the object [12–14]. Some end-to-end unsupervised learning methods [9] also output a semantic label that does not correspond to the actual category. The specific category to which the object belongs still depends on the clustering [15, 16]. 2) Due to the lack of manually injected supervised information, unsupervised learning split the data on their own. To verify, as shown in Fig. 1, we create a classification task of circles, triangles, and pentagons. The left samples are given color to the border, while the right samples are filled in the entire graph. We expect the unsupervised classification in these two different datasets to yield circles, triangles, and

---

[†] T. Han and J. Gao are co-first authors of the paper.
[*] Q. Wang is the corresponding author.

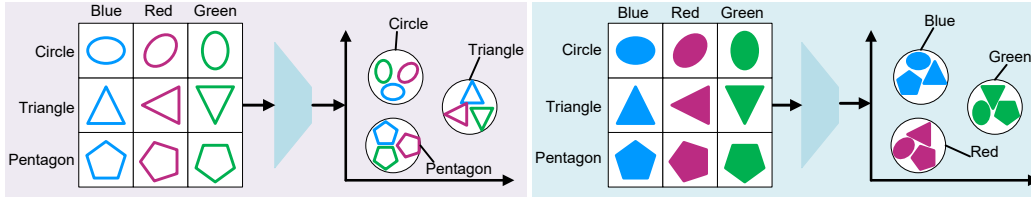

Figure 1: Two simple experiments on unsupervised classification with IIC [9]. Both of which we expect would yield shape-based categorization results. However, the data on the right, with the interference of color information, deviated far from our expectations.

pentagons. Nevertheless, the box on the left gives us a result based on shape, and the box on the right gives us a result based on color. The results are not expected. So it demonstrates that the features extracted by unsupervised learning with no supervised information are not all useful in promoting the task, even in some cases deviate far from the expectation. 3) Pure unsupervised learning is still difficult to deal with complex classification tasks. Taking the excellent IIC [9] as an example, it can achieve 99.2% accuracy in the simple handwritten numeric dataset MNIST [17], but it can only achieve 25.7% accuracy on the CIFAR100-20 [18].

To extract the beneficial feature and ignore the useless information in unsupervised learning, we make a constraint by injecting a part of the supervised information into unsupervised learning. In other words, this paper aims to establish an SSL framework combining unlabeled data with few labeled data. Unlike the existing excellent SSL methods based on consistent regularization and entropy minimization [19–22], we propose a new SSL framework via unsupervised semantic aggregation and deformable template matching. Specifically, the unlabeled data are explored to make self-supervised learning by maximizing mutual information. Then, few annotated samples are provided to align the semantic labels with the real categories. Besides, for further leveraging the unlabeled data, we establish a dynamic pattern pool for each class and assign proxy labels by template matching in the feature-level. It is a new and more reasonable pseudo label generation method that compares with other semi-supervised methods.

For problem 1), such a framework can eliminate the clustering operations that the traditional unsupervised learning required. For problem 2), the introduced supervised information can be an effective measure that helps unsupervised networks learn the expected representation under the interference of useless information (e.g., background, color). For question 3), maximum mutual information applied in semi-supervision learning will further enhance the classification performance on the complex datasets. Our main contributions are the following:

- Exploit triplet mutual information loss to achieve semantic labels clustering for unlabeled data in SSL, which has a better performance in unsupervised semantic aggregation than single paired MI loss.

- Propose a deformable template matching method for generating pseudo labels, which assigns proxy labels for unlabeled data in the high-level feature space. It is a more effective way compared with confidence based methods.

- Experiments on several standard classification benchmarks demonstrate that USADTM achieves state-of-the-art by integrating the supervised learning with unsupervised learning.

## 2 Related work

In this section, we briefly review the related unsupervised and semi-supervised learning in the classification task. More concrete introductions are provided in [23, 24].

**Unsupervised Learning**. Unsupervised learning focuses on learning the deep representation of unlabeled data [25–27]. AutoEncoder (AE) is a kind of neural network for unsupervised data representation [28–31]. The conventional process is training an autoencoder to compress and reduce dimensionality. Furthermore, classification learning or cluster algorithm is carried out based on the compressed feature of a middle layer. Variational Autoencoder (VAE), as a generative variant of AE, enforces the latent code of AE to follow a predefined distribution [32, 33]. Then Generative

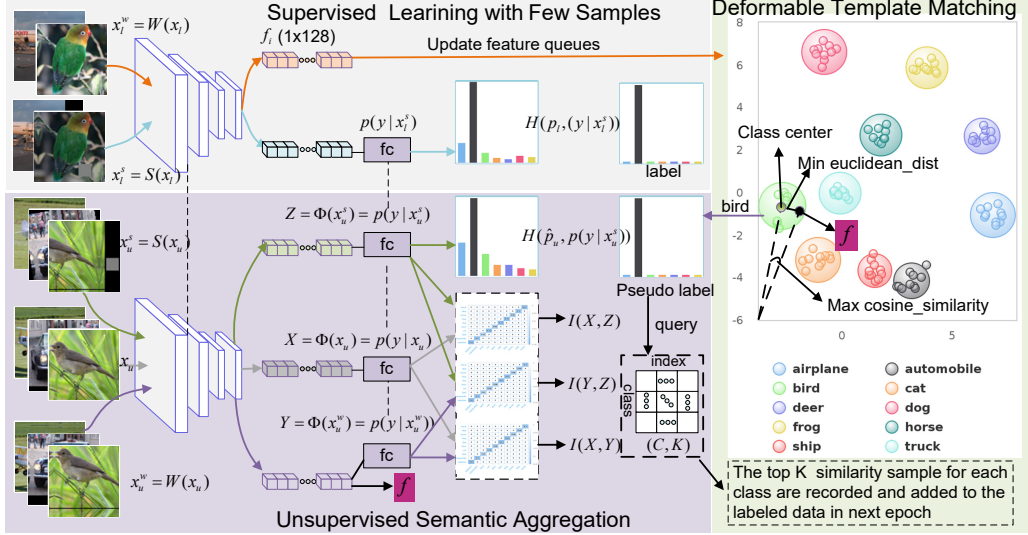

Figure 2: The flowchart of our proposed USADTM, which mainly consists of three components: 1) Supervised learning is trained with the labeled data; 2) A triplet mutual information loss is designed to generate the semantic labels for unlabeled data; 3) A label guesser based on feature-level deformable template matching is used to generate proxy labels. The dotted lines between the model represent shared parameters.

Adversarial Network (GAN) is another popular deep generative model in recent years [13, 34, 35]. Although the above generative learning has achieved excellent performance in some simple datasets (e.g., MINST [17]), it is still elusive for the generative model to achieve acceptable results in some complex datasets (e.g., CIFAR10 [18], CIFAR100 [18]). Another significant branch of unsupervised feature learning is based on information theory, which can extract cluster features by maximizing mutual information objective function [36, 12, 9, 37]. Notable is the IIC [9], which demonstrates the complete unsupervised SOTA results achieved on MNIST, CIFAR10, CIFAR100-20, and STL10 [38]. Unlike other works that focus on mutual information, it no longer estimates the mutual information between input and output, but directly train an end-to-end unsupervised classification model by maximizing the mutual information of paired samples on the output.

**Semi-supervised Learning**. Semi-supervised learning is a mixture of unsupervised and supervised learning [39]. Earlier work focused on consistency regularization [40, 41], Π-model [40] combines the supervised cross-entropy (CE) loss and the unsupervised consistency loss MSE. The former refers to the loss of labeled data, while the latter constrains the prediction of an unlabelled sample and its randomly augmented sample. Mean teacher [41] is based on the Π-model and Temporal Ensembling [40]. It uses an exponential moving average of model parameters to create predictions. Pseudo-labeling is a powerful technique for the unlabeled data entropy minimization in recent work. [42] is an early approach for estimating labels of unknown data. MixMatch [19] proposes a novel sharping method over multiple weak augmentation samples predictions to improve the quality of the Pseudo-Labels, L2 loss is used for unlabeled loss. UDA [21] proposes a augment scheme by combining AutoAugment [43] with Cutout [44] to generate the pseudo labels. The unsupervised loss is the Kullback Leiber divergence (KL). ReMixMatch [20] also uses weak augmentation to get pseudo labels. Then take the multiple predictions with strong augmentation to participate in the unlabeled loss calculation. FixMatch [22], a substantially simplified version of UDA and ReMixMatch, uses a confidence threshold to generate the pseudo labels. As far as we know, it is also the best semi-supervised framework at present.

## 3 Framework

In this section, we describe the SSL framework in detail. First, a triple mutual information maximization strategy is proposed to extract the semantic information for unlabeled data. Unlike the previous semi-supervised network [19, 20, 45, 21, 22], we then present a new pseudo-labeling approach based

on feature template matching, which assigns proxy labels to unlabeled data for entropy minimization. Finally, we introduce the object function for optimizing the proposed SSL framework.

## 3.1 Unsupervised Semantic Aggregation

For a semi-supervised classification task, we define it as follows: the training set has $\mathcal{C}$ categories and $\mathcal{X}$ samples. Then $\mathcal{X}$ is divided into the unlabeled set $\mathcal{X}_u$ and labeled set $\mathcal{X}_l$. Let $\{x_l^w, x_l^s\} \in \mathcal{X}_l$ are paired samples with the same one-hot labels $p_l$, and $\{x_u, x_u^w, x_u^s\} \in \mathcal{X}_u$ are triple paired unlabeled samples. Here, $x_*$ refer to the original images without any transform. Following [22], we also use two image enhancement strategies. $x_*^w$ and $x_*^s$ are converted data from the original image based on the weak augment $\mathcal{W}(\cdot)$ and strong augment $\mathcal{S}(\cdot)$, respectively. $\mathcal{W}(\cdot)$ is a standard flip-and-shift strategy. $\mathcal{S}(\cdot)$ is the RandAugment [46] strategy and followed by Cutout [44].

Feature extraction is an essential and basic task in unsupervised learning. A useful feature vector is to distinguish the sample from the whole dataset, that is, to extract the information belonging to the example. Mutual information fits this expectation well. The mutual information measures the KL divergence of the marginal distribution and joint distribution. If $(X, Y) \sim p(x, y)$, the mutual information between random variables $X$ and $Y$ is defined as follows:

$$
\begin{aligned}
\mathcal{I}(X; Y) &\equiv \mathrm{KL}(p(x, y) \| p(x)p(y)) \\
&= \sum_{y \in \mathcal{Y}} \sum_{x \in \mathcal{X}} p(x, y) \log\left(\frac{p(x, y)}{p(x)p(y)}\right) \\
&= H(X) - H(X|Y),
\end{aligned}
\tag{1}
$$

where the first row defines mutual information, the second row shows how to calculate mutual information in practical applications, and the third row reveals the relationship between mutual information and entropy and conditional entropy. IIC [9] regards the above mutual information as a criterion and directly maximizes it for paired samples to achieve automatically cluster for unlabeled data. However, single paired mutual information loss is challenging to deal with complex datasets. IIC is hard to optimize when the transformation used to generate the paired data is too strong. Therefore, we add weak augment data to serve as a transition. Every two groups among the three types of data are wrapped to obtain Triplet Mutual Information (T-MI) loss functions.

As illustrate in Fig. 2, the standard Wide ResNet-28-2 [47] is the classification network $\phi(\cdot)$. Let the output value of the last layer is $y$. $\phi(x_u) = p(y|x_u)$, $\phi(x_u^w) = p(y|x_u^w)$ and $\phi(x_u^s) = p(y|x_u^s)$ are the predicted class distribution produced by the model for input $x_u, x_u^w$ and $x_u^s$, respectively. We define triple mutual information loss as follows:

$$
\mathcal{L}_{T-MI}^u(x_u, x_u^w, x_u^s) = -\frac{1}{3}(\mathcal{I}(\phi(x_u); \phi(x_u^w)) + \mathcal{I}(\phi(x_u); \phi(x_u^s)) + \mathcal{I}(\phi(x_u^w); \phi(x_u^s))),
\tag{2}
$$

where each item is calculated according to the Eq. 1. For a batch, the marginal distribution probability $p(x), p(y)$, and the joint distribution probability $p(x, y)$ is based on the prediction results. The specific calculation method is flowing [9]. This maximum mutual information can make unlabeled data aggregation in semantic level, and clustering data on its own. Most importantly, T-MI makes the optimization become stable and a little bit simple. Next, we introduce the supervised learning to align semantic labels with labeled categories.

## 3.2 Deformable Template Matching

The annotated are exploited to play two roles. One is to align the clustering labels learned from mutual information with the labeled one-hot-labels, making end-to-end category predictions of unlabeled data. This purpose is realized by minimizing the cross-entropy loss of annotated data. The other is that with the mutual information loss, the unlabeled data's features will aggregate in feature space. A template matching algorithm can be conducted to assign proxy labels for unlabeled data by exploiting the feature distribution of labeled data in the final pooling layer. We call this Deformable Template Matching (DTM) because the feature distribution of labeled data is optimized continuously.

As shown in Fig. 2, the predictions of $x_l^s$ are used to minimize the cross-entropy. In the meanwhile, the weak augment data $x_l^w$ are used to generate template centers. Specifically, define a template feature pool where the feature of each class produced by the average pooling is stored as a queue

$\boldsymbol{Q}_i = \{\boldsymbol{f}_i^1, \boldsymbol{f}_i^2, \cdots, \boldsymbol{f}_i^N\}, (i \in [1, 2, \cdots, \mathcal{C}])$. $\boldsymbol{f}_i^j$ is a vector of length $1 \times 128$ (according to the Wide ResNet-28-2 [47]). Queue length $N$ is generally set up to $5\times$ length of per class labeled data. The new entered feature will edge out the end of the element when the queue is full. For each queue update, the template center are recalculated as below:

$$\boldsymbol{c}_i = \bar{\boldsymbol{Q}}_i = \sum_{j=1}^{N} \boldsymbol{f}_i^j, \tag{3}$$

where $\boldsymbol{c}_i (i \in [1, 2, \cdots, \mathcal{C}])$ are the clustering centers of $\mathcal{C}$ categories. Then, we match the unlabeled weak augment data to clustering centers. Given the feature vector $\boldsymbol{f}(1 \times 128)$ of an unlabeled sample, we calculate two metrics to determine the proxy label of the unlabeled data: cosine similarity and Euclidean distance. The index corresponding to the maximum cosine similarity and the minimum Euclidean distance is respectively calculated as follows:

$$m = \arg\max_i \{\frac{\boldsymbol{f}\boldsymbol{c}_i^T}{\|\boldsymbol{f}\|_2 \|\boldsymbol{c}_i\|_2}\}, \qquad n = \arg\min_i \{\sqrt{(\boldsymbol{f} - \boldsymbol{c}_i)(\boldsymbol{f} - \boldsymbol{c}_i)^T}\}, \tag{4}$$

where $m, n$ represent the class index. We calculate the two similarities to ensure that they are closest to the cluster center in both direction and distance. Finally, for an unlabeled image $x_i$, we define the following rules to assign labels:

$$\hat{p}_u^i = \begin{cases} m & (m = n \quad and \quad sim(\boldsymbol{f}, \boldsymbol{c}_m) \geq \tau) \\ -1 & (m \neq n \quad or \quad sim(\boldsymbol{f}, \boldsymbol{c}_m) < \tau)) \end{cases}, \tag{5}$$

where $-1$ is the ignored label, which indicates the sample will not be included in calculating loss. $sim(\boldsymbol{f}, \boldsymbol{c}_m)$ represents the cosine similarity between the under matched vector $\boldsymbol{f}$ and the potential cluster center $\boldsymbol{c}_m$. The purpose of setting the threshold $\tau$ is to filter out part of the wrong allocation. In the absence of special instructions, $\tau$ is generally set at 0.85. In each iteration, the proxy label is assigned to the unlabeled data using the proposed method above.

Besides, the available annotation data is often limited (e.g., four samples per class in CIFAR-10 [18], CIFAR-100 [18], and SVHN [48]). When a few examples are used to represent the category center, there is a deviation. To makes the cluster center more representative, as shown in the dashed box of Fig. 2, in each epoch, we set up a memory bank with $K$ lengths for each category, where samples in each class with the top $K$ confidence will be recorded and added to the labeled set in the next epoch. Memory bank is a matrix of $C \times K$ that stores the image ID and confidence. For a unlabeled sample, we query the corresponding row according to the predicted category. If the query sample has higher confidence than the lowest sample in the existing confidence values, then the query sample will replace it. In the experiment, the maximum value of $K$ follows the rules:

$$K = len(\mathcal{X}_l)/\mathcal{C} * 2. \tag{6}$$

Finally, there are two sets of annotated data, namely precisely labeled set $\mathcal{X}_l$ and fake labeled set $\hat{\mathcal{X}}_l$, where $\mathcal{C} \times K$ samples are selected from $\hat{\mathcal{X}}_l$ to enrich the samples of the feature pool. Noting that a large $K$ would make the class center too dependent on the additional data. It will create a negative effect when the $K$ samples contain some wrong predictions. Eq. 6 defines the $K$ based on our experiment. That is, the additional samples are controlled at two times of the labeled data. In the range of $K$ that we set up, the proposed method is robust. For a batch that $(x_l^s, p_l) \in \mathcal{X}_l$ and $(x_u^s, \hat{p}_u) \in \mathcal{X}_u$ with $N$ annotated samples, we minimized the cross-entropy of annotated and unannotated data:

$$\mathcal{L}_{CE}^l = \frac{1}{N} \sum H(p_l, p(y|x_l^s)), \qquad \mathcal{L}_{CE}^u = \frac{1}{N} \sum H(\hat{p}_u, p(y|x_u^s)). \tag{7}$$

## 3.3 Objective Function

In section 3.1 and 3.2, we define the unsupervised loss $\mathcal{L}_{MI}^u$ functions, supervised loss $\mathcal{L}_{CE}^u$ and $\mathcal{L}_{CE}^l$. The training of the proposed SSL framework is to minimize a weighted combination loss function of the three. The final objective function is summed as:

$$\mathcal{L}(x_l^s, x_u, x_u^w, x_u^s) = \mathcal{L}_{CE}^l + \mathcal{L}_{CE}^u + \alpha \mathcal{L}_{T-MI}^u, \tag{8}$$

where $\alpha$ denotes the weighting parameter of mutual information loss. Since T-MI loss is designed to assist the network in clustering, a large $\alpha$ will result in strong automatic clustering, which is not conducive to the alignment of cluster labels with actual categories. The recommended setting is 0.1.

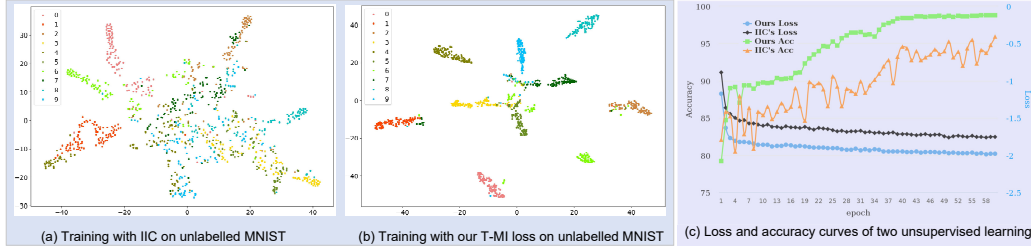

(a) Training with IIC on unlabelled MNIST     (b) Training with our T-MI loss on unlabelled MNIST     (c) Loss and accuracy curves of two unsupervised learning

Figure 3: Unsupervised learning on MNIST by using IIC and our proposed T-MI loss. The cluster distributions in (a) and (b) are drawn with the last average pooling layer 's feature of 1,000 samples (100 samples per class). To vivid show this cluster distribution, TSNE is used to make a dimensionality reduction. (c) shows the variation of training loss and test set accuracy for the two unsupervised losses under the same codebase.

## 4 Implementation Details

**Training Details**.    Following the previous work [19, 21, 21, 22], we use a standard Wide ResNet-28-2 [47] architectural as the model. During the training phase, the setting of batch size for labeled data and unlabeled data follows [22]. The labeled data and the unlabeled data and their transformations are input in parallelly. The SGD algorithm with $0.03$ initialization learning rate is adopted to optimize the network. The training and evaluation are performed on NVIDIA GTX 1080Ti GPU. To make the training process smoother, T-MI loss will not join the training until after some epochs. More specific settings are visible in the provided code.

**Testing Details**. Fllowing the previous work [41, 19–22], all of the datasets are evaluated on the test set, we also use the exponential moving average (EMA) of the model parameters when performing evaluation. In training phase, it is not adopted.

## 5 Experiments

We conduct our experiments from the following aspects: 1) For the proposed T-MI loss, we compare it with the single mutual information loss in IIC by performing unsupervised learning on the MNIST dataset (section 5.2). 2) Compare the performance of our DTM with other label guessers in generating proxy label (section 5.3). 3) To test the effectiveness of USADTM, we conduct the experiments on four standard SSL benchmarks (section 5.4). 4) An ablation study are performed to verify the contribution of each of USADTS's components (section 5.5).

### 5.1 Datasets

**CIFAR-10 and CIFAR-100** [18] are large datasets of tiny RGB images with size 32x32. Both datasets contain 60,000 images belonging to 10 or 100 classes respectively. Both sets provide 50,000 training labels and 10,000 validation labels.

**STL-10** [38] is dataset designed for unsupervised and semi-supervised learning. It only consists of 5,000 training labels and 8,000 valdiation labels. However, 100,000 unlabeled example image are also provided. These unlabeled examples belong to the training classes and some different classes. The images are 96x96 color images.

**SVHN** [48] Dateset is derived from Google Street View House Number.The data set contains the train, test, and the extra folder. It contains 33402, 13068 and 202353 labeled samples respectively. All Numbers have been adjusted to a fixed 32 x 32 resolution.

### 5.2 Comparison of Mutual Information Loss
Since our unsupervised loss is the inheritance and development of IIC [9], we are supposed to compare the triplet mutual information loss proposed in this paper with the unsupervised loss of IIC. The experimental configuration is as follows: 1) The codebase and hyperparameters configuration is completely consistent. The difference lies in the loss function. IIC used only the original image and its transformation as a pair, that is, the $\mathcal{I}(X, Z)$ in Fig. 2. 2) To abandon the clustering operation of

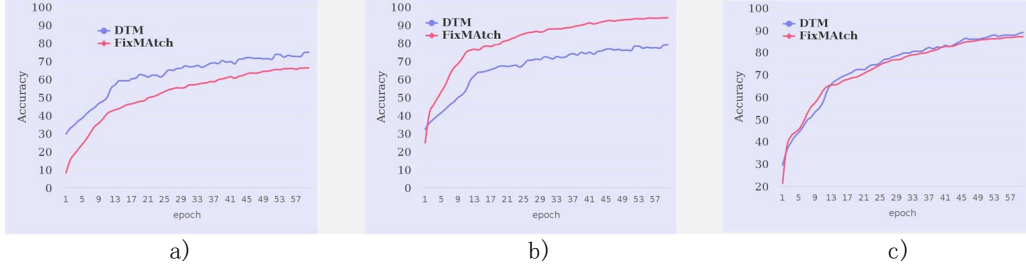

Figure 4: The comparison of label guessers in ours(DTM) and FixMatch. a) is the accuracy of valid labels over the whole unlabeled data. b) is the accuracy of the valid labels over the valid labels. c) is the accuracy of test set with the two label guessers.

IIC, we use 49,000 training samples from the MNIST training set for unsupervised learning. The remaining 1000 samples are exploited to determine the corresponding relationship between clustering labels and actual categories. It is not involved in network training. The training duration is 60 epochs.

The experimental results are shown in Fig. 3. Both Fig. 3 (a) and Fig. 3 (b) are from the model of 60 epoch. For the clustering performance, the T-MI loss achieves better feature differentiation. Fig. 3 (c) shows the training process. In terms of loss variation, T-MI converges faster and makes lower loss value. From the perspective of testing accuracy, T-MI performs more stable while IIC fluctuates greatly. The reason is that the weak data augmentation is introduced to constitute three pairs of MI loss. So it is more robust than single paired MI loss. The best T-MI result on the test set is 98.82%, and the IIC is 95.93% in this experiment.

## 5.3 Comparison of Proxy Label Generator

To demonstrate that our deformable template matching is more conducive to pseudo-label generation, we compared it with the current state-of-the-art FixMatch's label guesser. In this experiment, we transplant our DTM to a third-party PyTorch version of the FixMatch framework [49]. To make the comparison more comprehensive, we test the performance of the two methods in 40, 250, and 4,000 labeled samples.

Table 1: The accuracy for DTM and the Fix-Match's confidence based method.

| Label Generator | 40 labels | 250 labels | 4000 labels |
|---|---|---|---|
| FixMatch | 88.51 | 94.16 | 94.5 |
| DTM(ours) | **89.48** | **94.49** | **95.2** |

The experimental results are shown in Table 1, and it can be seen that our DTM achieves higher accuracy under the same FixMatch framework. To look for internal mechanisms, in Fig. 4, we plot the variation of the pseudo label's precision in the training phase for different generators. According to Fig. 4 a), DTM can achieve a higher correct proportion over the whole unlabeled set, while FixMatch has higher accuracy with valid labels, as shown in Fig. 4 b). Overall, DTM achieves a lower error rate in the test set than Fixmatch, as shown in Fig. 4 c). DTM outperforms FixMatch because the feature matching can capture more hard samples than confidence based methods.

## 5.4 Comparison of methods

We compare USADTM and other semi-supervised methods on the following four datasets: CIFAR-10, CIFAR -100, SVHN, and STL-10. Considering that FixMatch reimplements all the methods on the same codebase, we also use the same network architecture (Wide ResNet-28-2) as FixMatch to make the comparison as fair as possible. Besides, the optimizer's parameters, data preprocessing, and learning rate schedule are the same as FixMatch. The selection of unlabeled data also follows FixMatch. But on STL10, we added two additional sets of 40 labels and 250 labels. The results of the experiment are reported in Table 2. Our reports are the mean and variance value of the five times training under different random seeds. As the table shows, the proposed SSL framework produces state-of-the-art results in most of the settings. Taking CIFAR-10 and STL-10 as examples, we achieved an accuracy of 90.46% and 90.37% at the cost of four labeled samples in each category.

Table 2: Error rates for CIFAR-10, CIFAR-100, SVHN and STL-10 on different baseline models (Π-Model [40], Pseudo-Labeling [42], Mean Teacher [41], MixMatch [19], UDA [21], ReMixMatch [20], FixMatch [22]) and our proposed USADTM.

| | CIFAR-10 | | | CIFAR-100 | | | SVHN | | |
|---|---|---|---|---|---|---|---|---|---|
| Method | 40 labels | 250 labels | 4000 labels | 400 labels | 2500 labels | 10000 labels | 40 labels | 250 labels | 1000 labels |
| Π -Model | - | 54.26±3.97 | 14.01±0.38 | - | 57.25±0.48 | 37.88±0.11 | - | 18.96±1.92 | 7.54±0.36 |
| Pseudo-Labeling | - | 49.78±0.43 | 16.09±0.28 | - | 57.38±0.46 | 36.21±0.19 | - | 20.21±1.09 | 9.94±0.61 |
| Mean Teacher | - | 32.32±2.30 | 9.19±0.19 | - | 53.91±0.57 | 35.83±0.24 | - | 3.57±0.11 | 3.42±0.07 |
| MixMatch | 47.54±11.50 | 11.05±0.86 | 6.42±0.10 | 67.61±1.32 | 39.94±0.37 | 28.31±0.33 | 42.55±14.53 | 3.98±0.23 | 3.50±0.28 |
| UDA | 29.05±5.93 | 8.82±1.08 | 4.88±0.18 | 59.28±0.88 | 33.13±0.22 | 24.50±0.25 | 52.63±20.51 | 5.69±2.76 | 2.46±0.24 |
| ReMixMatch | 19.10±9.64 | 5.44±0.05 | 4.72±0.13 | 44.28±2.06 | 27.43±0.31 | 23.03±0.56 | 3.34±0.20 | 2.92±0.48 | 2.65±0.08 |
| FixMatch (RA) | 13.81±3.37 | 5.07±0.65 | **4.26**±0.05 | 48.85±1.75 | 28.29±0.11 | 22.60±0.12 | 3.96±2.17 | 2.48±0.38 | 2.28±0.11 |
| FixMatch (CTA) | 11.39±3.35 | 5.07±0.33 | 4.31±0.15 | 49.95±3.01 | 28.64±0.24 | 23.18±0.11 | 7.65±7.65 | 2.64±0.64 | 2.36±0.19 |
| Supervised(ours) | 74.92±4.73 | 44.48±0.57 | 16.03±0.38 | 86.07±1.14 | 61.03±0.50 | 38.81±0.21 | 86.60±2.65 | 51.68±1.65 | 14.44±0.98 |
| USADTM (ours) | **9.54**±1.04 | **4.80**±0.32 | 4.40±0.15 | **43.36**±1.89 | **28.11**±0.21 | **21.35**±0.17 | **3.01**±1.97 | **2.11**±0.65 | **1.96**±0.05 |
| Fully Supervised | 2.74 | | | 16.84 | | | 1.48 | | |

| | | STL-10 | | | | | |
|---|---|---|---|---|---|---|---|
| Method | 1000 labels | Method | 1000 labels | Method | 40 labels | 250 labels | 1000 labels |
| Π -Model | 26.23±0.82 | UDA | 7.66±0.56 | Supervised(ours) | 62.80±3.05 | 46.40 ±1.56 | 28.89±0.84 |
| Pseudo-Labeling | 27.99±0.80 | ReMixMatch | 5.23±0.45 | USADTM (ours) | **9.63**±1.35 | **6.85**±1.09 | **4.01**±0.59 |
| Mean Teacher | 21.43±2.39 | FixMatch (RA) | 7.98±1.50 | Fully Supervised | 1.48 | | |
| MixMatch | 10.41±0.61 | FixMatch (CTA) | 5.17±0.63 | | | | |

## 5.5 Ablation Study

USADTM consists of unsupervised learning and supervised learning with few labeled data. Unsupervised learning is the core of this paper, which includes T-MI loss and proxy label generator DTM. In this section, we study the impact of these components in the entire SSL framework by taking the CIFAR-10 dataset as an example. Explicitly, we define some comparative experiments by removing some components or changing some hyper-parameters.

The results in table 3 show that the USA and DTM both contribute to USADTM's performance. With the integration of the USA and DTM, it achieves a more comparable result. Comparing Exp3 and Exp7, we can find the DTM makes significant progress for unsupervised learning both in the 250 and 4000 labels settings. The comparison between Exp4 and Exp7 shows that the additional $K$ feature samples can help build a more reasonable feature center. Compared with Exp5∼8, the experimental performance is the best when $\tau$ is set at 0.85. The last two experiments and Exp7 are about the weight of T-MI loss. It can be found that the larger value will have a negative impact, while the smaller value has no apparent effect. So in other experiments, It is fixed as 0.1.

Table 3: Accuracy of ablation study on CIFAR-10 with 250 and 4000 labels. ($\alpha$ is the weight of T-MI loss, $\tau$ is cosine similarity threshold. Without special labeling, $\alpha$ is 0.1, and $\tau$ is 0.85 )

| Ablation | describe | 250 labels | 4000 labels |
|---|---|---|---|
| Exp1:Labeled(baseline) | Supervised learning with few samples | 45.52 | 83.97 |
| Exp2:Labeled+DTM | Add the deformable template matching to baseline | 94.27 | 95.01 |
| Exp3:Labeled+USA | Add unsupervised semantic (T-MI) to baseline | 81.23 | 88.34 |
| Exp4:1Labeled+USADTM ($K \equiv 0$) | $C \times K$ samples are not added in the SSL framework | 94.58 | 95.22 |
| Exp5:Labeled+USADTM ($\alpha = 0.1, \tau = 0.95$) | A complete SSL framework with $\alpha = 0.1, \tau = 0.95$ | 93.45 | 94.49 |
| Exp6:Labeled+USADTM ($\alpha = 0.1, \tau = 0.90$) | A complete SSL framework with $\alpha = 0.1, \tau = 0.90$ | 94.02 | 95.16 |
| Exp7:Labeled+USADTM ($\alpha = 0.1, \tau = 0.85$) | A complete SSL framework with $\alpha = 0.1, \tau = 0.85$ | **95.21** | **95.74** |
| Exp8:Labeled+USADTM ($\alpha = 0.1, \tau = 0.80$) | A complete SSL framework with $\alpha = 0.1, \tau = 0.80$ | 95.02 | 95.45 |
| Exp9:Labeled+USADTM ($\alpha = 1, \tau = 0.85$) | A complete SSL framework with $\alpha = 1, \tau = 0.85$ | 93.68 | 94.07 |
| Exp10:Labeled+USADTM ($\alpha = 0.01, \tau = 0.85$) | A complete SSL framework with $\alpha = 0.01, \tau = 0.85$ | 94.86 | 95.14 |

## 6 Conclusion

In this paper, we explore unsupervised learning in a semi-supervised classification task. A new triplet mutual information loss is proposed for unlabeled data's semantic aggregation, which is more

stable and effective than the single paired realization. Besides, we propose a feature-level deformable template matching to generate proxy labels aiming to improve the accuracy in existing pseudo-label generators. Experiments show that the new generator can capture more unlabeled samples for cross-entropy minimization. On several standard semi-supervised benchmarks, our proposed USADTM achieves the best performance on the whole at present, especially with a small number of labeled samples.

## Broader Impact

This work has the following potential positive impacts on the computer vision community. a) It contributes to the development of semi-supervised learning in combining the unsupervised and supervised learning. b) It can provide reference significances for various tasks related to image classification. Examples include but are not limited to medical image classification, scene classification, garbage classification, etc. c)This work may inspire other domains' research in the future. This work will not raise ethical problems for the foreseeable time.

## Acknowledgements

This work was supported by the National Natural Science Foundation of China under Grant U1864204, 61773316, 61632018, and 61825603.

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
