[Reviews · NeurIPS 2020]

Review 1

Summary and Contributions: In the paper, the authors propose a semi-supervised learning framework via unsupervised semantic aggregation and deformable template matching. Specifically, unsupervised semantic aggregation is used to generate semantic labels for unlabeled data by minimizing triplet mutual information, and then a few annotated samples and the feature-level deformable template matching are used to assign proxy labels to unlabeled data for entropy minimization. I have read the rebuttal and would keep the score.

Strengths: 1. The original image with its two transforms are formulated as the triple mutual information loss for feature extraction in unsupervised learning. The idea seems novel and technical sound. 2. The paper is clear and logical. The theoretical grounding is soundness and extensive experiments are conducted to validate the proposed method. 3. The work is related to the NeurIPS community.

Weaknesses: 1. The previous related works are discussed by authors. However, how this work differs from them are not clearly discussed. 3. In experiments section, the authors listed the experimental results and elaborated them. However, there is a lack of discussions and analyses w.r.t. them, e.g., in comparisons of mutual information loss, why T-MI performs more stable than IIC? In comparisons of proxy label generator, what characteristics make DTM outperform FixMatch? etc. In other words, the reason and rationality of the results shoud be discussed and analyzed.

Correctness: Yes.

Clarity: Yes

Relation to Prior Work: More discussions are expected.

Reproducibility: Yes

Additional Feedback: 1. See the weaknesses part. 2. Why the selection of K should follows the rules in Eq.(6)? How different K impact the performance of the proposed method? 3. In Section 4, testing details, ``Fllowing" -> Following.


Review 2

Summary and Contributions: This paper addresses the problem of classification under a few annotated samples by injecting a part of the supervised information into unsupervised learning. Specifically, the authors combine both unsupervised learning and semi-supervised learning (SSL) to propose an Unsupervised Semantic Aggregation and Deformable Template Matching (USADTM) framework for SSL. For the proposed method, unsupervised semantic aggregation based on Triplet Mutual Information (T-MI) minimization is explored to generate semantic labels for unlabeled data. Then the semantic labels are aligned to the actual class by the supervision of labeled data. The proposed deformable template matching method for generating pseudo labels is a more effective way compared with confidence-based methods. Comprehensive experiments are conducted to demonstrate the effectiveness of the proposed methods.

Strengths: 1. It is novel to generate proxy labels based on deformable template matching comparing with the existing semi-supervised methods. 2. On the whole, the paper is well written, and the proposed semi-supervised framework is novel and easy to follow. 3. The proposed Triplet Mutual Information (T-MI) minimization function is used to evaluate unsupervised learning effectively, which could address the challenging difficult evaluation of unsupervised learning. Furthermore, the proposed T-MI loss function has a more robust and stable performance comparing with the original single paired mutual information functions. 4. The experimental results are comprehensive and convincing, which is comprehensively demonstrate the effectiveness of the proposed method.

Weaknesses: However, I still have some concerns about the paper as follows: 1. Do the two CNNs share the same parameters in Fig. 2? If it is true, it is better to give some annotations to explain this in the figure. 2. In Lines 169-176, the authors mentioned that there is a deviation when limited available annotated data (e.g., four samples per class) are used to represent the category center. Thus, in each epoch the authors set up a memory bank with K lengths for each class, where samples in each class with the top K cosine similarity to the class center will be recorded and added to the labeled set in the next epoch. Is the choice of Eq. (6) based on experiments or theoretical analysis? Moreover, is the proposed method robust to the selected unlabeled samples? Will the proposed method select many wrong samples? 3. How to update the memory banks? 4. What the difference and relation between the proposed method and contrastive learning? For example, momentum Contrast for Unsupervised Visual Representation Learning.

Correctness: Yes, the claims and method are correct. Yes, the empirical methodology is correct.

Clarity: Yes, this paper is well written and easy to follow.

Relation to Prior Work: Yes, the difference between this work and the previous works is well discussed in the paper.

Reproducibility: Yes

Additional Feedback:


Review 3

Summary and Contributions: This paper proposes an Unsupervised Semantic Aggregation and Deformable Template Matching (USADTM) framework for SSL semi-supervised learning (SSL), which combines the unsupervised learning and semi-supervised learning to improve the performance and reduces the cost of data annotation. The USA produces the semantic label for unlabeled data by optimizing a Triplet Mutual Information loss. And DTM generates the pseudo labels for unlabeled samples in leveraging few samples. Experimental results compared with eight state-of-the-art methods on four datasets demonstrate the validity of the proposed approach. Overall, I think the paper is slightly above the borderline.

Strengths: 1. This paper is novel, original, and well structured. It clearly references existing work on supervised and semi-supervised learning and gives an excellent overview of the area. 2. The proposed triplet mutual information loss achieves semantic label clustering for unlabeled data and has a better performance in unsupervised semantic aggregation than single paired MI loss according to the comparison experiment in section 5.2. 3. The proposed deformable template matching method for generating proxy labels is a new perspective than the current works. According to section 5.3, it achieves comparable or even better results than other methods. 4. Detailed design and analysis of ablation experiments reveal the effectiveness of each component proposed in the paper. In particular, the improvement brought by the fake labels is quite apparent.

Weaknesses: 1. It is not fully clear how the top K similarity for each class is selected to help generates more reasonable class centers. Elaborating this mechanism would help the reader. 2. The authors only give the K value selection with a class number of 10 and 100, How to confirm the K when the class number is others? 3. In the Fig. 2, the , and fc need more explanations. 4. On page 6, line 242, MNIST needs citation.

Correctness: Yes

Clarity: Yes

Relation to Prior Work: Yes

Reproducibility: Yes

Additional Feedback:


Review 4

Summary and Contributions: The authors combine both to propose an unsupervised semantic aggregation and deformable template matching framework for semi-supervised learning, which strives to improve the model's performance while reducing the cost of data annotation

Strengths: (1) The author claim that exploit triplet mutual information loss to achieve semantic labels clustering for unlabeled data in semi-supervised learning. (2) The author claim that propose a deformable template matching method for generating pseudo labels.

Weaknesses: 1. It seems trivial to extend the Triplet Mutual Information [1] and its code [2]. The contribution of the proposed method is not clear. Please explain the difference between your work and [1] about Triplet Mutual Information. [1] Wu, Jianlong, et al. "Deep comprehensive correlation mining for image clustering." Proceedings of the IEEE International Conference on Computer Vision. 2019. [2] https://github.com/Cory-M/DCCM 2. Are these parameters the same for different tasks in Table 3? Is it possible to do a 2D search instead of fixing one parameter while searching for another? 3. For the comparison, how were the parameters of other methods tuned? 4. Deformable template matching is an existing technology. Please explain the difference between your work and [3, 4] separately. [3] Lee, Hyungtae, et al. "DTM: Deformable template matching." 2016 IEEE International Conference on Acoustics, Speech and Signal Processing (ICASSP). IEEE, 2016. [4] Xu, Yuhao, et al. "Partial descriptor update and isolated point avoidance based template update for high frame rate and ultra-low delay deformation matching." 2018 24th International Conference on Pattern Recognition (ICPR). IEEE, 2018.

Correctness: Just.

Clarity: Just

Relation to Prior Work: Yes.

Reproducibility: No

Additional Feedback:

[Author Response · NeurIPS 2020]

We thank the reviewers for their positive and constructive feedbacks of this work. Now, a detailed broader impact
analysis is already included in our current draft. Then, we address the comments as follows.

# 1    Common comments

**R1(Additional)**: Why K should follows the Eq. (6)? How different K impact the performance? **R2(Q2)**: Is Eq. (6)
based on experiments or theory? Is it robust for different K? **R3(Q1)**: a further explanation for K would help the reader.
**R**: In our framework, the samples with top K confidence in each class will be added as template samples in the next
epoch. So a large K would make the class center too dependent on the additional data. It will create a negative effect
when the K samples contain some wrong predictions. Eq. (6) defines the K based on our experiment. That is, the
additional samples are controlled at two times of the labeled data. In the range of K that we set up, the proposed method
is robust. Besides, we will further elaborate on this mechanism in the revision according to the reviewers' comments.

# 2    Response for R1: Thanks for the helpful comments.

**Q1**: How this work differs from related works? **R**: The differences are: 1) Design T-MI loss to achieve semantic labels
clustering for unlabeled data compared to other SSL frameworks. 2) Generate proxy labels from the feature space while
others rely on the prediction scores. We will clarify the differences more clearly in the revision.

**Q2**: Why T-MI performs more stable and what characteristics make DTM outperforms FixMatch should be discussed.
**R**: T-MI loss performs more stable because the weak data augmentation is introduced to constitute three pairs of MI loss.
So it is more robust than single pair MI loss. DTM outperforms FixMatch because the feature matching can capture
more hard samples than confidence based methods. We will add more analyses for the two comparison experiments.

# 3    Response for R2: Thanks for the careful consideration.

**Q1**: Do the two CNNs share the same parameters in Fig. 2? **R**: Yes, we will note it in the revision.

**Q2**: How to update the memory bank? **R**: Memory bank is a matrix of $C \times K$ that stores the image ID and confidence.
For a unlabeled sample, we query the corresponding row according to the predicted category. If the query sample has
higher confidence than the lowest sample in the existing confidence values, then the query sample will replace it.

**Q4**: What the difference and relation between the proposed method and contrastive learning [1]?
**R**:1) **Difference**: [1] learns a classifier for each sample. Unsupervised training is achieved by identifying instances
with contrastive loss, while ours is achieved by maximizing T-MI loss of the sample and its two augmentation images.
They are two different methods. 2) **Relation**: we both use a queue to store past samples, but our method only saves the
features of the labeled and the $C \times K$ additional samples. We will also include this discussion in the revision.

# 4    Response for R3: Thanks for the positive comments.

**Q1,3,4**: How to confirm K when class number is not 10 or 100? The tiny question in Fig. 2, the citation of MNIST.
**R**:According to our experiment, Eq. (6) is applicable when the number of categories is not 10 or 100. We will add an
explanation for fc in the revised version. MNIST is now cited in the current draft.

# 5    Response for R4: Thanks for the carefully checking.

**Q1**: It seems trivial to extend the Triplet Mutual Information and its code [2]. Please explain the difference in TMI. **Q4**:
Please explain the difference of Deformable Template Matching (DTM) between your work and [3, 4] separately.

**R**: We clarify that our work is not to extend the [2] and its code. We are sorry that the T-MI and DTM are named the same
as the previous works, so it confuses the reviewer. In fact, they are different in tasks, structures, and implementations,
and the codes are developed by ourselves. **For T-MI**, the differences between ours and [2] are: 1 ) TMI in [2] measures
the MI between a sample and its positive and negative samples. Our T-MI loss evaluates the MI among the original
image and its weakly and strongly augmentation images. 2) In [2], the feature map is used to calculate the MI while we
use the prediction scores in the output layer to calculate the MI. 3) MI in [2] is obtained indirectly through discriminator
while we use a simple and effective calculation method based on statistics. **For DTM,** 1) DTM in [3] is achieved by a
set of pre-defined basic rules. 2) In [4], the new template is generated by updating the template descriptor and adding
new keypoints with the matching process in pixels. 3) Our DTM refers to updating the class centers in the training
phase. They are completely different implementations.

**Q2**: Are Table 3's parameters the same for different tasks? Is it possible to do a 2D search for hyper-parameters?
**R**: Yes, we select a best parameter group ($\alpha = 0.1, \tau = 0.85$) according to the ablation experiments in Table 3. Then,
the parameter group is fixed on other tasks. The reason we fix one parameter and search another is that our computation
resource is limited. We will add a figure to show the results of the 2D search in the revised version.

**Q3**: For the comparison, how were the parameters of other methods tuned?
**R**: The results of the other methods shown in the paper are under the best parameters in the original papers. They select
the best hyper-parameters by a series of comparative experiments. We follow the same selection strategy.

**Reference**
[1] He, Kaiming, et al. "Momentum contrast for unsupervised visual representation learning," in CVPR, 2020.
[2]Wu, Jianlong, et al. "Deep comprehensive correlation mining for image clustering," ICCV, 2019.
[3] Lee, Hyungtae, et al. "DTM: Deformable template matching," ICASSP, 2016.
[4] Xu, Yuhao, et al. "Partial descriptor update and isolated point avoidance based template update for high frame rate and ultra-low delay deformation matching," ICPR, 2018.


[Meta-Review · NeurIPS 2020]

The reviewers found the usage of the deformable templates here interesting and this work will be useful to a large part of the NeurIPS community.